# Targeting Redox Metabolism in Pancreatic Cancer

**DOI:** 10.3390/ijms22041534

**Published:** 2021-02-03

**Authors:** Nadine Abdel Hadi, Gabriela Reyes-Castellanos, Alice Carrier

**Affiliations:** Centre de Recherche en Cancérologie de Marseille (CRCM), CNRS, INSERM, Institut Paoli-Calmettes, Aix Marseille Université, F-13009 Marseille, France; nadine.abdelhadi@inserm.fr (N.A.H.); gabriela.reyes-castellanos@inserm.fr (G.R.-C.)

**Keywords:** pancreatic ductal adenocarcinoma, cancer metabolism, mitochondria, mitochondrial metabolism, redox metabolism, ROS, oxidative stress, cancer therapeutic strategy

## Abstract

Cell metabolism is reprogrammed in cancer cells to meet their high bioenergetics and biosynthetic demands. This metabolic reprogramming is accompanied by alterations in redox metabolism, characterized by accumulation of reactive oxygen species (ROS). Elevated production of ROS, mostly by mitochondrial respiration, is counteracted by higher production of antioxidant defenses (mainly glutathione and antioxidant enzymes). Cancer cells are adapted to a high concentration of ROS, which contributes to tumorigenesis, metastasis formation, resistance to therapy and relapse. Frequent genetic alterations observed in pancreatic ductal adenocarcinoma (PDAC) affect KRAS and p53 proteins, which have a role in ROS production and control, respectively. These observations led to the proposal of the use of antioxidants to prevent PDAC development and relapse. In this review, we focus on the therapeutic strategies to further increase ROS level to induce PDAC cell death. Combining the promotion of ROS production and inhibition of antioxidant capacity is a promising avenue for pancreatic cancer therapy in the clinic.

## 1. Introduction

The most frequent pancreatic cancer is pancreatic ductal adenocarcinoma (PDAC). PDAC remains a poor-outcome disease, with the lowest 5-year survival rate after diagnosis (currently 9%) [1,2]. As a result that this cancer is often asymptomatic, most patients are diagnosed at an advanced or metastatic state, when the tumor cells gained resistance to chemotherapeutic drugs [3,4]. Unlike many other cancers, PDAC incidence and death rates have been rising over the past decade [2]. There is thus an imperative need to develop effective therapeutic strategies against this incurable disease.

In that context, promising therapeutic strategies are devoted to target metabolic dependencies of PDAC. The knowledge of many features of metabolic reprogramming in PDAC did greatly increase during the past decade [5,6,7,8,9]. In particular, it is now known that mitochondrial metabolism is essential in PDAC as in other cancers, co-existing with aerobic glycolysis [10,11]. Mitochondria are multifunctional organelles involved in the balance between cell life and death. Their main functions are the production of energy on the form of ATP by the respiratory chain in the inner membrane, the production of reactive oxygen species (ROS) also by the respiratory chain, and the intrinsic pathway of apoptosis. Their central role in the metabolism of carbohydrates, proteins and lipids is opening therapeutic options that we recently reviewed [11]. In the present review, we will focus on the role of mitochondria in ROS control, and on the approaches for targeting redox metabolism in PDAC.

## 2. Pancreatic Cancer

Pancreatic ductal adenocarcinoma (PDAC) represents 85% of all pancreatic tumors. It remains a mortal disease with dismal prognosis and a rising incidence. PDAC is the fourth-leading cause of cancer deaths in United States of America [2], and the 13th worldwide [1]. A major concern is that PDAC is predicted to become the second cause of cancer death by 2030 in USA [2,12,13]. Importantly, pancreatic cancer belongs to the cancers with the lowest 5-year relative survival rate (9%), pointing out a very slight increase of this survival from 2.5% (1970–1977) to only 8% (2007–2013) (https://ourworldindata.org/cancer). This situation reflects the slow progress against this disease due to a lack of early diagnosis and major treatment advances.

PDAC is a malignant epithelial neoplasia with glandular (ductal) differentiation that arises from the exocrine portion of the pancreas and is mainly found in the head of the organ [14]. PDAC can be histologically graded according to its glandular differentiation [14]. More recently, PDAC tumors were divided in two transcriptional subtypes: the ‘classical’ and a more aggressive ‘squamous’/‘basal-like subtype’, in which cancer cells fail to undergo proper differentiation [15]. Importantly, pancreatic cancer evolves from multistage progression of precursor lesions known as pancreatic intraepithelial neoplasias (PanINs) and intraductal papillary mucinous neoplasms (IPMNs). These lesions are relevant because they suggest the possibility to detect and treat them before an incurable cancer develops [14].

PDAC is characterized by its aggressiveness and intrinsic resistance to conventional chemotherapy. This is probably due in part to its characteristic tissular architecture: cancer cells are surrounded by a dense stroma (desmoplasia) composed of activated fibroblasts, immune cells, nervous tissue, and a high amount of collagen fibers [16]. This abundant stroma compresses the vasculature, which limits oxygen, nutrients, and drug delivery to the cells [17,18]. However, even if this stroma is a source of both physical and oxidative stress, PDAC cells are well adapted to such hostile conditions to reach their energetic and biosynthetic demands (i.e., by reprogramming their metabolism) [9,11].

The major genetic event in PDAC is an activating point mutation of the KRAS oncogene (70–95%), being permanently activated, and maintaining cancer proliferation, transformation, invasion, and survival [19]. Oncogenic KRAS mutations are an early event in low grade PanINs, and subsequent mutations in the tumor suppressor genes TP53, CDKN2A, and SMAD4 occur later during disease progression and are also frequently observed. Unfortunately, targeting mutated KRAS in the clinic has been unsuccessful, and for most of patients, cytotoxic chemotherapy will continue to be the standard of care [3]. 

PDAC is a silent disease that lacks symptoms at an early stage. In consequence, pancreatic cancer is usually diagnosed at a late stage and 85% of the patients present with an unresectable tumor and/or metastatic disease at diagnosis. In these patients, chemotherapeutic regimens with gemcitabine, nab-paclitaxel plus gemcitabine, 5-fluorouracil, leucovorin, irinotecan, and oxaliplatin (FOLFIRINOX), or radiotherapy, provide only modest increase in survival [3,19,20,21]. In contrast, for those rare patients able to undergo complete surgical resection (10–15%), the standard of care with the modified FOLFIRINOX (with a modification to exclude bolus doses of 5-fluorouracil) has provided the longest median survival (54 months) [3,22]. Unfortunately, the majority of these patients eventually relapse and die, suggesting that most of them presented undetectable micrometastases at the time of resection [23].

In conclusion, the aggressive nature and specific features of PDAC justify the need to explore alternative and novel therapeutic approaches. In this context, the essential role of mitochondrial metabolism in pancreatic cancer has been demonstrated [10,11,24,25,26,27,28]. Therefore, targeting mitochondrial metabolism, in particular redox metabolism, is an attractive therapeutic option.

## 3. ROS and Cancer

Reactive oxygen species (ROS) are oxygen-derived molecules, both reactive free radicals and nonradical molecules (Figure 1). Free radicals are mostly superoxide anion radicals (O_2_^−^) and hydroxyl radicals (HO). The main nonradical molecule is hydrogen peroxide (H_2_O_2_) which is stable and has a well-established role in cellular signaling [29]. The mitochondrial respiratory chain producing ATP by oxidative phosphorylation (OXPHOS) and the membrane-bound NADPH oxidases (NOXs) are the two major intracellular sources of ROS in both normal and cancer cells (Figure 1) [29,30]. ROS can also be generated as by-products of certain biochemical reactions such as prostaglandin synthesis, detoxification reactions by cytochrome P450, and β-oxidation in peroxisomes [31].

The cellular ROS level is tightly controlled through a battery of antioxidant defenses, both enzymatic and nonenzymatic (small molecules) antioxidants (Figure 1). Oxidative stress is a condition that occurs when there is an imbalance between ROS generation and antioxidant response, leading to an excess of ROS [29,32]. ROS—mostly the free radicals superoxide anion O_2_^−.^ and hydroxyl radical HO^.^—oxidize cell macromolecules (nucleic acids, proteins, lipids, and carbohydrates), contributing to the disruption of cell functions. By causing oxidative DNA damage, genomic instability, and modifying gene expression, the overproduction of ROS promotes cancer development, chemoresistance, and relapse [33,34,35]. Alteration of the redox metabolism is a hallmark of cancer cells, tightly linked to the global metabolic reprogramming [36,37,38]. Cancer cells show an adaptation to metabolic alterations sustaining tumorigenesis, resulting in elevated ROS production which is counterbalanced by a high rate of ROS scavengers’ production.

### 3.1. ROS as Signaling Molecules in Cancer

In cancer, ROS play an important role in regulating many aspects of cell processes, from signaling, proliferation, and survival, to promotion of oxidative damage and cell death [36]. Hence, ROS have a dual role in cancer depending mostly on their intracellular concentration [39]. 

An increased production of ROS has been shown to promote several pro-tumorigenic cell signaling and cell survival pathways. ROS regulate the phosphatidylinositol-3 kinase (PI3K)/protein kinase B (Akt) and mitogen activated-protein kinase (MAPK)/extracellular-regulated kinase 1/2 (ERK1/2) pathways [40]. It has been shown in many cancers that ROS induce the activation of PI3K/Akt pathway by oxidizing and inactivating the phosphatase and tensin homolog (PTEN), protein tyrosine phosphatase 1B (PTP1B), and protein phosphatase 2 (PP2) [37,40]. Moreover, several proteins such as RAS and RAF have been shown to stimulate the MAPK/ERK signaling pathway and induce cancer cell proliferation [41]. For example, ROS generated from mitochondria are required for oncogenic KRAS-driven lung cancer growth resulting from MAPK/ERK activation [30]. High ROS levels can also promote tumor progression and aggressiveness by the activation of redox-sensitive nuclear factor kappa-light-chain-enhancer of activated B cells (NF-κB) and the master regulator of antioxidant response Nrf2 (see below), thus achieving resistance to drug therapies [40,41]. Furthermore, some studies have reported that cancer cells undergo metabolic changes that include the activation of AMP-activated protein kinase (AMPK) to enhance NADPH production and maintain redox balance [42].

While mild-to-moderate levels of ROS are associated with the activation of pro-tumorigenic survival and growth pathways, excessive concentration of ROS can lead to the induction of cell cycle arrest and cell death [29,39]. We will develop this context more deeply below.

### 3.2. Mitochondrial ROS and Oxidative Stress

Mitochondria are a major source of ROS in normal and cancer cells, since ROS are by-products of the oxidative phosphorylation (OXPHOS) in the respiratory chain producing ATP. The respiratory chain is composed of 5 protein complexes: Complexes I to IV are electron transport thain (ETC) complexes using oxygen as the last acceptor of electrons at Complex IV, and the last complex (Complex V) is the ATP synthase. Mitochondrial superoxide anion (O_2_^−.^) is generated by partial reduction of oxygen mostly at the level of Complexes I and III. Depending on the context, increased production of mitochondrial ROS (mtROS) can result from either decreased or increased OXPHOS. On the one hand, mitochondrial dysfunction caused by an aberrant mitochondrial ETC has been implicated in decreased OXPHOS [43] and increased production of intracellular ROS and oxidative stress [44]. High ROS production associated with mtDNA mutations may support metabolic reprogramming, participate in cancer cell behavior determination, and promote chemotherapy resistance [31,38]. On the other hand, mtROS up-regulation in response to an oncogenic insult may be mediated by OXPHOS. For example, a recent study in acute myeloid leukemia demonstrated that cancer cells with pre-existing and persisting chemoresistance display: (a) high levels of ROS; (b) increased mitochondrial mass; (c) a high OXPHOS gene signature; and (d) high OXPHOS activity [45]. In addition, in ovary cancer, high ROS level is associated with high OXPHOS activity [46]. Moreover, tumor biopsies from melanoma patients with disease progression and melanoma cell lines with acquired drug resistance demonstrate preexisting high expression of mitochondrial biogenesis genes. Targeting mitochondrial biogenesis proteins overcomes drug resistance in a subset of cell lines via the PI3K/Akt-mediated mTORC1 signaling pathway [47]. Altogether, there is a body of evidence showing the pivotal role of mtROS generation either by decreased or increased OXPHOS in cancer cells.

### 3.3. Evasion of Mitochondrial ROS through Antioxidant Defenses in Cancer

Cells have developed several antioxidant systems to maintain an appropriate level of ROS and prevent their accumulation. The production of superoxide anion (O_2_^−.^) by mitochondria is countered by the superoxide dismutases (SODs) that catalyze the conversion of O_2_^−.^ to hydrogen peroxide (H_2_O_2)_. The glutathione peroxidases (GPXs), peroxiredoxins (PRXs), and catalase (CAT) enzymatically facilitate the detoxification of H_2_O_2_ [48]. Small antioxidant molecules (glutathione (GSH), NADPH, vitamins A, C, and E) also play a crucial role in the antioxidant defenses. In some cancers, the antioxidant enzymes SODs, GPXs, and CAT are found to be overexpressed [49]. The relative effect of each of these enzymes on scavenging H_2_O_2_ in a particular cell type may depend on their expression levels, their compartmental localization, and other redox-active components that modify their function [50]. Importantly, GSH and NADPH are overproduced by reprogrammed metabolism in cancer cells [36]. 

Glutathione is the most abundant antioxidant molecule in the cells. Cancer cells have increased glutathione levels to alleviate the effects of oxidative stress [51]. The antioxidant function of glutathione is mediated by two enzymes: GSH peroxidase (GPX) and glutathione disulfide (GSSG) reductase (GR). GPX allows the reduction of H_2_O_2_ by the oxidation of GSH (reduced glutathione, biologically active form) to GSSG (oxidized glutathione). The GSSG is subsequently reduced to GSH by GR at the expense of NADPH used as a cofactor [52,53]. The GSH/GSSG ratio is often used as a convenient expression of the cellular redox status. Thus, glutathione synthesis and NADPH production actively support H_2_O_2_ detoxification [54]. Moreover, it has been established that GSH and GSH-related enzymes are a major contributing factor to drug resistance [51,55].

During the oxidative stress response, cancer cells up-regulate the expression of some transcriptional factors that mitigate oxidative stress. The nuclear factor erythroid 2-related factor 2 (Nrf2) is the master regulator of the cellular antioxidant response and a prime target of research in cancer prevention and treatment. Nrf2 abundance within the cell is tightly regulated by kelch-like ECH-associated protein 1 (KEAP1), a redox-sensitive E3 ubiquitin ligase substrate adaptor. In response to oxidative stress, KEAP1 is oxidized at reactive cysteine residues, resulting in its dissociation from Nrf2. Nrf2 then translocates into the nucleus and binds to the antioxidant response element (ARE) sequence in the promoter of its target genes (Figure 1) [56]. It has been shown that Nrf2 controls the expression of key components of the glutathione (GSH) and thioredoxin (TXN) antioxidant systems, as well as enzymes involved in NADPH regeneration, ROS, and heme metabolism, thus playing a fundamental role in maintaining the redox homeostasis of the cell [57]. Furthermore, a recent study showed that restoring redox homeostasis through Nrf2 activation is imperative for tumor recurrence [58], in agreement with other studies demonstrating the importance of antioxidant pathways for acquired resistance to targeted therapies in cancer [59].

### 3.4. ROS in Cancer: The Achilles’ Heel of Cancer Cells?

As cancer cells have elevated ROS levels compared with normal cells, they are potentially more vulnerable to oxidative stress-induced cell death. Interestingly, several chemotherapeutics drugs have been shown to promote anti-tumorigenic effect by significantly increasing ROS production and thus causing irreversible oxidative damage (listed in Table 1) [38,57]. Further increasing ROS level in cancer cells can also be attained through decreasing antioxidant defenses. For example, GSH depletion seems to have a profound effect on cell survival and chemosensitivity by promoting oxidative stress-induced cancer cell death [51]. Current strategies include inhibition of GSH synthesis by targeting the enzyme γ-glutamylcysteine ligase (GCL) or by interfering with the uptake of cystine (precursor of cysteine) through inhibition of the xCT (also known as x_c_^−^ or SLC7A11) antiporter system [60,61]. In many studies, it has been shown that compounds inhibiting de novo GSH synthesis, such as buthionine sulphoximine (BSO), NOV-002 and sulphasalazine, exhibit anticancer activity by strongly increasing ROS levels (Table 1) [62,63]. Moreover, co-treatment with BSO and the anti-CAT compound arsenic trioxide has been shown as a new cancer treatment approach [64]. Aside from lowering GSH pools, another important branch of thiol metabolism has emerged as targets for the selective killing of cancer cells, including peroxiredoxins (PRDXs) and thioredoxins (TRXs) [61].

## 4. Targeting Mitochondrial Redox Metabolism in Pancreatic Cancer

In pancreatic cancer, the relationship between redox regulation and oncogenic transformation remains an intense topic of research, as it is increasingly clear that redox vulnerabilities may provide therapeutic opportunities. In fact, ROS have been shown to be critical for mutant oncogenic KRAS-driven transformation [65] and pancreatic tumor growth [9]. For example, superoxide anion (O_2_^−.^) is a prosurvival factor in PDAC. In addition, it has been shown that oncogenic KRAS induces an increase in ROS production mediated by mitochondrial dysfunction and NADPH oxidase activities alteration [66]. A major mediator of KRAS-mtROS signaling is the serine/threonine kinase Protein Kinase D1 (PKD1) (Figure 2) [67]. ROS-activated PKD1 promotes cell survival by inactivating c-Jun N-terminal kinase (JNK) 1/2 and p38 signaling, but also through the activation of NF-κB. PKD1 also promotes proliferation by up-regulating ERK1/2 and epidermal growth factor receptor (EGFR) signaling [68]. Other functions for ROS-activated PKD1 are up-regulation of inflammatory cytokines, regulation of autophagy, and chemoresistance [66]. Oncogenic KRAS up-regulates antioxidant defense systems via various mechanisms, counteracting the excessive ROS accumulation during pancreatic cancer progression. One mechanism is through the generation of NADPH (required for regeneration of reduced glutathione) by the enzymes malic enzyme 1 (ME1) and isocitrate dehydrogenase 1 (IDH1) [9,69]. Another mechanism is the activation of the transcription factor Nrf2, which was shown to be critical for KRAS-driven tumorigenesis in PDAC models (Figure 2) [70]. 

The tumor suppressor p53 is also frequently altered in PDAC by mutation or deletion. p53 has many antitumoral protective functions including its antioxidant ability [71]. In our laboratory, we characterized a target of p53, the Tumor Protein 53-Induced Nuclear Protein 1 (TP53INP1), as playing a crucial role in the antioxidant defense through regulating p53 activity in the nucleus and participating in the elimination of ROS-producing altered mitochondria by mitophagy in the cytoplasm [72,73]. TP53INP1 is lost in early stages of PDAC development before the acquisition of p53 mutations, and this loss can support the oncogenic KRAS-associated oxidative stress [74,75]. In addition, mutant p53 proteins can acquire so-called gain-of-function activities and influence the cellular redox balance in various ways, for instance by binding to the Nrf2 transcription factor, a major regulator of cellular redox state as discussed above [76].

Alleviating oxidative stress in PDAC clinically is a strategy both in prevention (to prevent PDAC initiation, promotion, and progression) as well as in therapy to dampen the toxic effects of chemotherapy and prevent tumor relapse [39,77]. Since PDAC has the highest incidence of KRAS mutation among all types of cancers, the inhibition of mutant KRAS-driven pathways could be a good strategy to prevent this cancer. One possibility is to use agents as anti-tumoral therapies, like vitamin E and vitamin C that inhibit the mutant KRAS-driven pathways via MAPK or PIK3/Akt and ERK1/2, and counteract oxidative stress [77]. Moreover, some studies show that using an ROS-related pathway blocker such as longikaurin E and nexrutine decreases ROS in PDAC and induces apoptosis by inhibiting p38-MAPK and STAT3/LC3, respectively (Figure 2) [39]. Another possibility is to administer mitochondrial-targeted antioxidants such as mitoQ, with the overall goal to suppress KRAS-induced mtROS formation. 

Nevertheless, in the therapeutic context, the most promising strategy is to increase intracellular ROS levels to make pancreatic cancer cells more vulnerable to oxidative stress-induced cell death. For example, compound 3b and benzyl isothiocyanate activate p38, JNK, and ERK leading to the induction of apoptosis mediated by enhanced ROS [78,79]. In addition, it has been reported that some drugs can induce ROS accumulation by enhancing mitochondrial dysfunction and loss of mitochondrial membrane potential, such as resveratrol (prooxidant at high doses) and spiclomazine [39]. Moreover, the use of arsenic trioxide combined with parthenolide was shown to induce ROS generation and apoptosis via the mitochondrial pathway in human pancreatic cancer cells (Figure 2) [80]. Similarly, a new compound from quinazolinediones (QDs) family (QD325) has been shown as the most potent redox modulator candidate for the treatment of PDAC [81]. QD325 induces Nrf2-mediated oxidative stress and blocks mitochondrial function (through inhibition of mtDNA transcription and downregulation of mtDNA-encoded OXPHOS enzymes), leading to substantial ROS production and pancreatic cancer cell death.

## 5. Conclusions

As observed for several other cancers, pancreatic cancer cells are endowed with an excessive accumulation of ROS. This supports the strategy of treating pancreatic tumors with drugs that increase ROS levels to reach a ROS concentration promoting cell death. Inducing ROS production together with the inhibition of antioxidant systems would be more efficient in promoting cell death. Thus, combining these two strategies is a promising avenue for pancreatic cancer therapy in the clinic (Table 1).

## Figures and Tables

**Figure 1 ijms-22-01534-f001:**
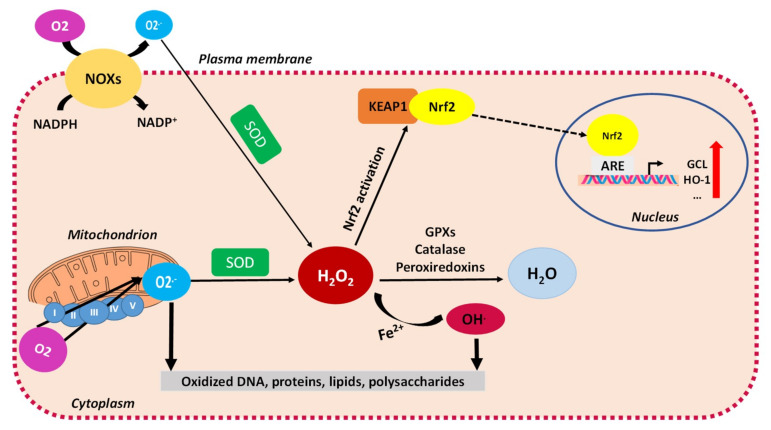
Schematic representation of ROS generation and enzymatic antioxidant defenses. Superoxide anion (O_2_^−.^) is mainly produced by NADPH oxidases (NOXs) and by the mitochondrial ETC complexes by partial reduction of molecular oxygen (O_2_). Superoxide is dismutated into hydrogen peroxide (H_2_O_2_) by superoxide dismutases (SODs). H_2_O_2_ is converted to water (H_2_O) by glutathione peroxidases (GPX), peroxiredoxins (PRX), or catalase. Via the Fenton reaction with metal ions Fe^2+^ or Cu^+^, H_2_O_2_ is further reduced to highly reactive hydroxyl radical (OH^.^), thereby damaging biological macromolecules such as DNA, lipids, and proteins. H_2_O_2_ is the main player in redox homeostasis, and can induce the activation of Nrf2 through dissociation of the Nrf2-KEAP1 complex, phosphorylation of Nrf2, and its nuclear translocation. In the nucleus, Nrf2 promotes transcription of antioxidant genes by binding to the ARE in the promoter region. Abbreviations: ARE, antioxidant response element; ETC, electron transport chain; GPXs, glutathione peroxidases; H_2_O_2_, hydrogen peroxide; HO-1, heme oxygenase-1; KEAP1, Kelch-like ECH-associated protein 1; NADPH, nicotinamide adenine dinucleotide phosphate hydrogen; NADP+, nicotinamide adenine dinucleotide phosphate; NOXs, NADPH oxidases; Nrf2, nuclear factor erythroid 2-related factor 2; O_2_, dioxygen; O_2_^−^, superoxide anion; OH, hydroxyl radical; SOD, superoxide dismutase; TXN, thioredoxin.

**Figure 2 ijms-22-01534-f002:**
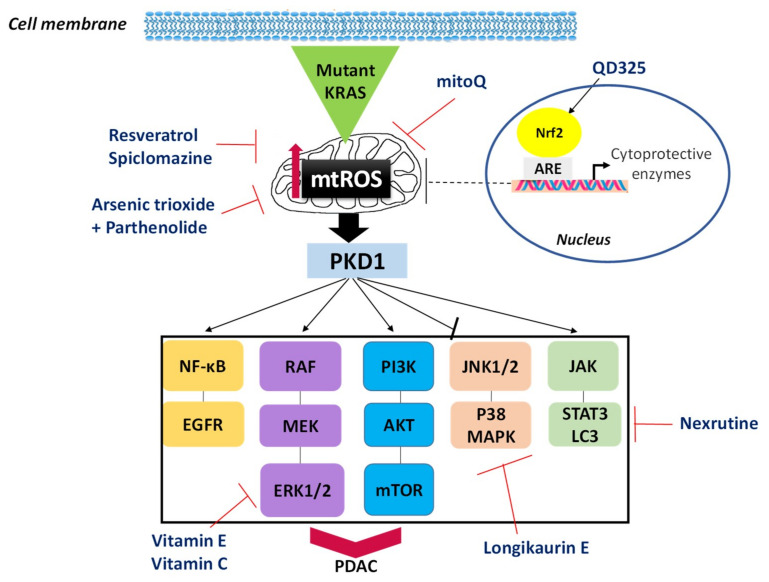
Mutant KRAS-driven signaling pathways via mitochondrial ROS-induced activation of protein kinase D1 (mtROS/PKD1), and potential options for therapeutic intervention in pancreatic cancer. Activation of PKD1 is mediated by increase in mtROS promoted by mutant Kras expression. ROS-activated PKD1 promotes cell survival by inactivating JNK1/2 and p38 signaling, and activating NF-κB and EGFR signaling. PKD1 also promotes proliferation by up-regulating ERK1/2 and PI3K. To prevent PDAC development, one possibility is to use antioxidant therapies like vitamins E and C, longikaurin E, nexrutine, and mitoQ. In a therapeutical context, some drugs can induce ROS accumulation and apoptosis such as resveratrol, spiclomazine, arsenic trioxide combined with parthenolide, and QD325 through Nrf2-mediated oxidative stress. Abbreviations: ARE, antioxidant response element; Akt, protein kinase B; EGFR, epidermal growth factor receptor; ERK1/2, extracellular-regulated kinase 1/2; JAK, Janus kinases; JNK 1/2, c-Jun N-terminal kinase 1/2; LC3, light chain 3; MAPK, mitogen activated-protein kinase; MEK, mitogen-activated protein kinase; mTOR, mechanistic (previously mammalian) target of rapamycine; mtROS, mitochondrial reactive oxygen species; NF-κB, nuclear factor kappa-light-chain-enhancer of activated B cells; Nrf2, nuclear factor erythroid 2-related factor 2; PDAC, pancreatic ductal adenocarcinoma; PI3K, phosphatidylinositol-3 kinase; PKD1, protein kinase D1; RAF, rapidly accelerated fibrosarcoma; STAT3, signal transducer and activator of transcription protein 3.

**Table 1 ijms-22-01534-t001:** Anticancer treatments regulating ROS levels.

Drugs	Mechanism of Action	Cancer Types	Context	Median Survival Rate	Ref(s)
**Chemotherapeutic drugs targeting the redox metabolism in cancer**					
Gemcitabine	DNA synthesis inhibition. Induces the accumulation of ROS and increases the capacity of antioxidant programs	Pancreatic cancer	In vitro, and in vivo		[39]
Taxanes (Paclitaxel and docetaxel)	Promote mitochondrial cell death through the release of cytochrome c Disrupt the mitochondrial electron transport chain	Different types of cancer	In vitro, and in vivo		[82,83]
Anthracyclines (Doxorubicin or epirubicin)	Insert into the DNA of replicating cells and inhibit topoisomerase II, which prevents DNA and RNA synthesis	Different types of cancer	Clinical	Metastatic Breast Cancer: 7–8 months	[84,85]
Arsenic trioxide (As_2_O_3_)	Impairs the function of the mitochondrial electron transport chain Inhibits GPx, TrxR and CAT	Acute promyelocytic leukemia and lung cancer	In vitro, in vivo, and clinical	Acute promyelocytic leukemia: complete remission	[31,32,86,87]
Methotrexate	Triggers ROS-associated cell apoptosis	Different types of cancer	In vitro		[88]
Mitoxantrone	Triggers cell membrane scrambling	Different types of cancer	In vitro		[89]
Tamoxifen	Promotes cancer cell senescence	Colon and breast cancer	In vitro		[90]
Cisplatin	Generation of nuclear DNA adducts	Different types of cancer	In vitro, and clinical	NSCLC: 9.1 months	[91,92,93]
ATN-224	Inhibits SOD1 Inhibits ETC complex IV	NSCLC and prostate cancer	In vivo, and clinical	Prostate cancer: median progression-free survival 30 weeks	[94,95]
**Compounds targeting the *de novo* GSH synthesis**					
Buthionine sulphoximine (BSO)	Inhibits GCL activity and de novo GSH synthesis Enhances A_2_O_3_ activity	Ovarian, breast and pancreatic cancer, melanoma	In vitro, and in vivo		[57,96,97]
NOV-002	Glutathione disulfide mimetic that alters the intracellular GSH/GSSG ratio	Lung, breast and ovarian cancer	Clinical	Advanced NSCLC ∼ 8.5 months	[98,99]
Sulphasalazine	Inhibitor of cysteine/glutamate antiporter xCT; reduces intracellular transport of cysteine required for GSH synthesis	Pancreatic and lung cancer	In vitro, and in vivo		[100,101]
L-asparaginase	Depletes glutamine, reduces GSH	Leukemia and pancreatic cancer	In vitro, in vivo, and clinical	PDAC: overall survival 6.0 months (combo with chemotherapy) versus 4.4 months (chemotherapy alone)	[102,103,104]
Erastin	Downregulates cysteine redox shuttle and blocks GSH regeneration	Different types of cancer	In vitro		[105,106]
(1S, 3R)-RSL3 (RSL3)	Induce ferroptosis without depleting the GSH pool	Lymphoma and renal carcinoma	In vivo		[107]

Abbreviations: CAT, catalase; ETC, electron transport chain; GPX, glutathione peroxidase; GSH, reduced glutathione; GSSG, glutathione disulfide; GCL, glutamate cysteine ligase; NSCLC, non-small-cell lung carcinoma; ROS, reactive oxygen species; SOD1, superoxide dismutase 1; TrxR, thioredoxin reductase.

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
