# Peer review of "Targeting Redox Metabolism in Pancreatic Cancer"

_ijms, 2021, doi:10.3390/ijms22041534_

Round 1

Reviewer 1 Report

In this MS, Authors Nadine Abdel Hadi and collaborators, proposed a review on oxidative stress and cell metabolism as preferential targets to fight pancreatic cancer development.

The MS deals about a subject of great interest and Authors chose Pancreatic Ductal AdenoCarcinoma (PDAC) as it represents the most frequent pancreatic cancer.

After a short introduction, MS is divided into three paragraphs is 15 pages long and cites more than one hundred recent publications. One table and two figures constitute nice elements in this interesting and very readable MS.

That said I have minor points to address:

  • Page 6 section 3.3. police size seems to be different in several part of this section.
  • Legend of figure 2, Authors wrote “In a therapeutical context, some drugs can induce ROS accumulation and apoptosis such as resveratrol…” I do not understand how an antioxidant such as resveratrol can induce ROS accumulation.
  • Do Authors have data on oxidative damages of protein in PDAC context? Might proteolytic systems such as proteasome represent putative target to fight PDAC?

Author Response

Reviewer 1

In this MS, Authors Nadine Abdel Hadi and collaborators, proposed a review on oxidative stress and cell metabolism as preferential targets to fight pancreatic cancer development.

The MS deals about a subject of great interest and Authors chose Pancreatic Ductal AdenoCarcinoma (PDAC) as it represents the most frequent pancreatic cancer.

After a short introduction, MS is divided into three paragraphs is 15 pages long and cites more than one hundred recent publications. One table and two figures constitute nice elements in this interesting and very readable MS.

Response: We warmly thank the Reviewer for appreciating our Review.

That said I have minor points to address:

  • Page 6 section 3.3. police size seems to be different in several part of this section.

Response: we did homogenize the police type and size.

  • Legend of figure 2, Authors wrote “In a therapeutical context, some drugs can induce ROS accumulation and apoptosis such as resveratrol…” I do not understand how an antioxidant such as resveratrol can induce ROS accumulation.

Response: We understand the issue raised by the Reviewer. Actually, many antioxidant molecules such as resveratrol and Vitamin C show antioxidant or prooxidant activity depending of the dose (for resveratrol, see for example this review: Madrigal-Perez L.A. and Ramos-Gomez M., IJMS 2016, Resveratrol Inhibition of Cellular Respiration: New Paradigm for an Old Mechanism). Resveratrol at high doses can induce apoptosis through mitochondrial ROS production increase (see for example Juan M. et al., J Agrig Food Chem 2008, Resveratrol Induces Apoptosis through ROS-dependent mitochondria pathway in HT-29 human colorectal carcinoma cells).

We added “(prooxidant at high doses)” in the sentence “In addition, it has been reported that some drugs can induce ROS accumulation by enhancing mitochondrial dysfunction and loss of mitochondrial membrane potential, such as resveratrol (prooxidant at high doses) and spiclomazine [39].” page 10.

  • Do Authors have data on oxidative damages of protein in PDAC context? Might proteolytic systems such as proteasome represent putative target to fight PDAC?

Response: Thank you for these two interesting questions.

For the first question, it was indeed reported in PDAC that cellular ROS can oxidize cysteine thiol groups, affecting the catalytic activity or conformation of the corresponding protein. See for example this nice paper (Chio I. I. C. et al., Cell 2016, NRF2 Promotes Tumor Maintenance by Modulating mRNA Translation in Pancreatic Cancer). We added this reference in the Table 1 and references list.

For the second question, inhibition of proteasome function has indeed emerged as a strategy for PDAC therapy. See these two papers for example:  Awasthi N. et al., HPB 2009, Proteasome inhibition enhances antitumour effects of gemcitabine in experimental pancreatic cancer; Lankes K. et al., Molecular Oncology 2020, Targeting the ubiquitin-proteasome system in a pancreatic cancer subtype with hyperactive MYC.

Reviewer 2 Report

The present Review is a very well-organized and written one. The topic is very interesting and merits further publication

Author Response

The present Review is a very well-organized and written one. The topic is very interesting and merits further publication.

Response: We warmly thank the Reviewer for appreciating our Review.

Reviewer 3 Report

I think the review was well written and covers all the relevant points for ROS targeted therapeutics for pancreatic cancer. 

Some pieces of advice for the authors:

shorten sentences that run for 3+ lines (like lines 37-41, 234-238, etc.)

Shorten the ROS and cancer section. the review is supposed to focus on pancreatic cancer specifically. I appreciate the background but I feel it could be more concise. I would suggest significantly shortening section 3.3

In table 1, if the clinical data available, id like to see a column listing median survival rates with ROS therapies. 

Author Response

I think the review was well written and covers all the relevant points for ROS targeted therapeutics for pancreatic cancer. 

Response: We warmly thank the Reviewer for appreciating our Review.

Some pieces of advice for the authors:

Shorten sentences that run for 3+ lines (like lines 37-41, 234-238, etc.)

Shorten the ROS and cancer section. the review is supposed to focus on pancreatic cancer specifically. I appreciate the background but I feel it could be more concise. I would suggest significantly shortening section 3.3.

Response: Thank you for these advices to improve our manuscript. We did shorten 3 sentences which were too long: pages 2, 3, 5. We did shorten the section 3.3. keeping the objective to maintain the maximum of information necessary to understand the following sections 3.4. and 4, and Table 1.

In table 1, if the clinical data available, I’d like to see a column listing median survival rates with ROS therapies. 

Response: This is a very good idea. We did add in the Table 1 a column listing survival rates and associated references. Please consider the revised Table 1 in additional word file. Text additions in this new column appear in red color character. The 6 additional references appear in red color character in the References section.